# Therapeutic Use and Chronic Abuse of CNS Stimulants and Anabolic Drugs

Daniela Coliță [1], Cezar-Ivan Coliță [1], Dirk M. Hermann [2], Eugen Coliță [1], Thorsten R. Doeppner [3,4], Ion Udristoiu [5,*] and Aurel Popa-Wagner [2,5,*]

1. Doctoral School, University of Medicine and Pharmacy "Carol Davila", 020276 Bucharest, Romania
2. Chair of Vascular Neurology, Dementia and Ageing, University Hospital Essen, University of Duisburg-Essen, 45147 Essen, Germany
3. Department of Neurology, University Medical Center Göttingen, 37075 Gottingen, Germany
4. Department of Neurology, University Hospital Giessen, 35394 Giessen, Germany
5. Department of Psychiatry, University of Medicine and Pharmacy of Craiova, 200349 Craiova, Romania
* Correspondence: ion.udristoiu@gmail.com (I.U.); aurel.popa-wagner@geriatrics-healthyageing.com (A.P.-W.)

**Abstract:** The available evidence suggests that affective disorders, such as depression and anxiety, increase risk for accelerated cognitive decline and late-life dementia in aging individuals. Behavioral neuropsychology studies also showed that cognitive decline is a central feature of aging impacting the quality of life. Motor deficits are common after traumatic brain injuries and stroke, affect subjective well-being, and are linked with reduced quality of life. Currently, restorative therapies that target the brain directly to restore cognitive and motor tasks in aging and disease are available. However, the very same drugs used for therapeutic purposes are employed by athletes as stimulants either to increase performance for fame and financial rewards or as recreational drugs. Unfortunately, most of these drugs have severe side effects and pose a serious threat to the health of athletes. The use of performance-enhancing drugs by children and teenagers has increased tremendously due to the decrease in the age of players in competitive sports and the availability of various stimulants in many forms and shapes. Thus, doping may cause serious health-threatening conditions including, infertility, subdural hematomas, liver and kidney dysfunction, peripheral edema, cardiac hypertrophy, myocardial ischemia, thrombosis, and cardiovascular disease. In this review, we focus on the impact of doping on psychopathological disorders, cognition, and depression. Occasionally, we also refer to chronic use of therapeutic drugs to increase physical performance and highlight the underlying mechanisms. We conclude that raising awareness on the health risks of doping in sport for all shall promote an increased awareness for healthy lifestyles across all generations.

**Keywords:** aging; depression; cognitive decline; dementia; traumatic brain injuries; stroke; restorative therapies; recreational drugs; doping





## 1. Introduction

Cerebrovascular and neurodegenerative diseases are a major cause of death and disability worldwide that increase in number each year. Given the severity of brain damage inflicted by traumatic brain injuries, stroke, dementia, or Parkinson disease, much research has been devoted to developing drugs with which to slow the disease progression and to improve recovery of the diseased brain. The efficacy of the available treatments is nevertheless limited. However, most of the same therapeutic drugs that are used daily in the clinic are increasingly used by children, teenagers, and athletes for recreational purposes or to increase physical performance in competitive sports. However, chronic use of brain stimulants may cause serious health-threatening conditions including, infertility, subdural hematomas, liver and kidney dysfunction, peripheral edema, cardiac hypertrophy, myocardial ischemia, thrombosis, and cardiovascular disease. In this review, we focus on the impact of chronic use of brain stimulants on psychopathological disorders, cognition,

and depression with the aim to emphasize the health risks of chronic overdose use of therapeutical drugs. In this review, we hope to increase the awareness for healthy lifestyles in young people. We also cover the mechanisms underlying the therapeutic efficacy of drugs currently in use in geriatric medicine and aging-associated brain diseases.

## 2. Therapeutic Approaches to Restore Cognitive and Motor Tasks in Brain Aging and Disease

### 2.1. Aging, Cognitive Decline, and Dementia of Alzheimer's Type

Behavioral neurology and neuropsychology studies show that cognitive decline is a central feature of aging impacting the quality of life [1]. Among elderly individuals, depressive symptoms are common in individuals over the age of 65, and 1 to 2% of individuals over the age of 65 meet the criteria for major depressive disorder [2]. It is estimated that, after the age of 65, combined cognitive decline and low mood doubles with every 5 years so that, by the age of 85, one in three individuals suffers from Alzheimer's disease (AD) [3]. Moreover, between 10 and 20% of individuals above the age of 65 are diagnosed with mild cognitive impairment (MCI) [4], and of these, 10% will progress to AD [5]. Moreover, depression is common and medically relevant illness that has been associated with a state of "accelerated aging" and can significantly compromise successful aging [6]. Indeed, a recent meta-analysis showed evidence for the association between affective disturbances and decline in non-specific cognitive function in older adults [6].

In recent years, the concept of "brain reserve" has emerged to describe some individuals having an increased inherent adaptive neuroplasticity thereby providing greater resilience to depressive behavior [7]. For example, during aging, compensatory mechanisms, and neuroplasticity may be at work in the brain. For example, the brains of aged individuals compensate for age-related decline by recruiting additional neural networks to maintain performance on the task requiring executive control [8,9].

Dementia of AD type is characterized by progressive deterioration of cognitive functioning due to various brain pathologies. For example, by the Mini-Mental State Examination, the temporal cortex, where pathological change initially starts, has been associated with the loss of learning/memory abilities in patients during early stage dementia [10]. More recently, neuroimaging techniques such as magnetoencephalography (MEG) reflect both learning/memory and executive functions and could be an independent biomarker of cognitive impairment [11]. This is important because psychiatric comorbidity is often a neglected confounding factor in studies of subjective cognitive complaints [12].

### 2.2. Pathophysiology of Cognitive Decline

Individuals 60 years or older are at increased risk for developing cognitive impairment caused by disruption of neuroanatomic regions during perioperative events (surgery itself or anesthesia) after cardiovascular diseases [13,14]. For example, memory functions have been classically associated with the medial temporal lobe including the hippocampus and the entorhinal cortex, the basal forebrain, and the thalamus. Executive memory on the other side, is associated with the frontal cortex, subcortical nuclei, and white matter fibers. Of note, at 3 months following noncardiac surgery, patients experienced cognitive decline, and individuals with executive function disturbance had the more severe functional limitations [15].

Traumatic brain injuries (TBI) are nondegenerative acquired brain injuries causing neurocognitive impairment, including persistent cognitive, emotional, behavioral, or somatic disability [16]. A large study recently has shown that processing speed particularly influences functional outcome in TBI subjects [16,17].

Cognitive impairment and memory dysfunction following stroke diagnosis are common among stroke survivors and impacts the quality of life. Stroke also affects the cognition including attention, memory, language, and orientation. Moreover, one third of stroke patients are at risk for vascular cognitive impairment (VCI) and develop dementia within 1 year of stroke onset [18]. However, the significance of time and etiologic stroke subtype on cognitive symptom profiles, course over time, and pathogenesis is not known in detail. Thus, a recent multicenter prospective cohort study showed that post-stroke cognitive impairment becomes manifested both early and long-term after stroke, while executive function and language improve over time [19].

### 2.3. Therapeutic Approaches to Restore Cognitive and Motor Tasks

#### 2.3.1. Acetylcholine Pathway

Loss of memory is a core feature of many neurological and psychiatric disorders including Alzheimer's disease and schizophrenia. Current treatment options for memory loss are very limited, and the search for safe and effective drug therapies has, until now, had limited success. Acetylcholine is released in the brain during learning and is critical for the acquisition of new memories. Until now, the only effective treatment to slow cognitive decline during aging or in diseases such as Alzheimer's, is using drugs that broadly prevent acetylcholine degradation. However, this leads to multiple adverse side effects. Therefore, the discovery of specific receptor targets that have the potential to provide the positive effects whilst avoiding the negative ones is promising. Recently, specific acetylcholine receptor targets that boost the positive effects whilst avoiding the negative ones have been identified [20]. The findings identified specific receptors for the neurotransmitter acetylcholine that re-route information flowing through memory circuits in the hippocampus.

Disruption of cholinergic function by TBI and stroke may produce cognitive and motor impairments [16,21]. Indeed, acetylcholinesterase inhibitors are promising drugs for the treatment of the post-stroke cognitive impairment. Indeed, a meta-analysis of placebo-controlled studies has shown that acetylcholinesterase inhibitors, most notably donepezil, a centrally selective cholinesterase inhibitor with relatively limited side effects, improve cognitive impairment following TBI and maintain a stable pattern of improved cognitive function in patients with vascular dementia and post stroke cognitive impairment [22–24].

#### 2.3.2. Catecholaminergic Pathway

The pathophysiology of TBI includes changes in the levels of catecholamines, dopamine (DA), epinephrine (EPH), and norepinephrine (NE) [25]. Dopaminergic drugs (e.g., methylphenidate, amantadine, and bromocriptine) are psychostimulants that may improve brain plasticity and some features of executive function but also arousal and the speed of information processing [16]. Dopamine (3,4-dihydroxyphenethylamine) is a monoamine neurotransmitter synthesized in the brain from L-DOPA that plays a major role in the motivational component of reward-motivated behavior but also in movement and cortical plasticity. Indeed, DA is a key mediator of motor skill learning [25–27].

DA stimulants include methylphenidate, amphetamine, and methamphetamine.

Methylphenidate may improve the rate of functional recovery, cognition, and attention during the acute rehabilitation training [28,29]. Therefore, methylphenidate is also recommended in the treatment of cognitive and motivational deficits [30]. Amphetamines, such as methamphetamine and 3,4-methylenedioxymethamphetamine (MDMA), belong to a class of compounds called phenethylamines with catecholaminergic effects in the central nervous system (CNS) and peripheral circulation. In the clinic, amphetamines are used to treat short-term obesity, narcolepsy, and attention deficit hyperactivity disorder

(ADHD). Methamphetamines, including dextroamphetamine (Adderall), are powerful stimulants of the nervous system that increase the release of DA and NE from presynaptic nerve terminals and is frequently used in the treatment of attention and memory impairment following TBI [31–33]. However, more recent studies report that chronic exposure to methamphetamines before brain trauma may accentuate the pathophysiological signs of injury, worsening TBI outcomes [34].

A recent study aiming at finding physiological correlates of the effects of dextroamphetamine on working-memory performance has been performed in healthy controls. Dextroamphetamine improved performance by increasing working-memory load that approached working-memory capacity albeit only in those subjects who had relatively low working-memory capacity at baseline [35].

Epinephrine (also called adrenaline) is a catecholamine produced by chromaffin cells within the adrenal medulla. The adrenaline agonist, phenylephrine, is an alpha-1 adrenergic drug which when given intravenously triggers vascular constriction and increases blood pressure. Therefore, it is utilized commonly in critical care for cardiovascular support. Indeed, vasopressors, including phenylephrine, are used to support blood pressure in TBI patients [36]. Catecholamines including NE and phenylephrine also seem to increase cerebral blood flow in various animal models and in patients [37]. Poststroke hypotension may cause neurological deterioration. Indeed, phenylephrine-induced hypertensive therapy successfully restored neurologic dysfunction caused by hypotension and stopped infarct progression in patients with ischemic penumbra [38–40].

The stimulant effects of caffeine on the CNS have been known for centuries. In the brain, caffeine, besides influencing cognitive performance, increases the perception of alertness and wakefulness. Indeed, caffeine consumption was associated with a significantly lower risk of developing neurodegenerative diseases, such as Alzheimer's and Parkinson's diseases [41–44]. At the cellular level, caffeine blocks adenosine receptors, mainly $A_1$ and $A_{2A}$ subtypes, causing an increased release of DA, noradrenalin, and glutamate [45].

Cocaine is the third most common illicit substance of abuse after cannabis and alcohol. Nevertheless, the U.S. Food and Drug Administration (FDA) has approved cocaine hydrochloride for pain control before minor nose, mouth, and throat surgery. Dentists or oral surgeons can also use cocaine as topical anesthetic before painful procedures.

Morphine is an opiate obtained from poppy seeds used to relieve moderate, severe, and chronic pain. It is also used for pain relief after major surgeries and treatment for cancer-related pain. Morphine is still routinely used today, though there are a number of semi-synthetic opioids of varying strength such as codeine, fentanyl, methadone, oxycodone, hydrocodone, hydromorphone, and meperidine. Heroin or diamorphine is an opiate made from morphine converted in the body to morphine. Heroin is a potent analgesic, five to ten times more potent than morphine. It is not available for clinical use in USA. However, it is available in Great Britain.

### 2.3.3. Serotonergic Drugs

Serotonin or 5-hydroxytryptamine (5-HT) is a monoamine neurotransmitter synthesized in serotonergic neurons of the CNS from tryptophan that plays a role in cognition, particularly response inhibition and memory consolidation, and modulates numerous physiological processes such as vomiting and vasoconstriction.

Motor deficits are common after stroke and affects subjective well-being and are linked with reduced quality of life. Currently, restorative therapies that target the brain directly are available [46,47]. Most of the drugs that are used to enhance motor recovery after stroke in humans are serotonergic and dopaminergic agents [48]. There are numerous studies which suggest the clinical utility of selective serotonin reuptake inhibitor (SSRI) drugs for promoting improved motor outcome after stroke [49,50]. Stroke causes long-term disability due to disruption of descending motor pathways causing significant sensory-motor deficits. Using a combination of electrical stimulation and functional connectivity mapping in a model of stroke, it was shown that administration of an old drug, gabapentin,

improves sensory-motor deficits and structural and functional plasticity of the corticospinal pathway [51].

A polygene score explains differences in L-DOPA effects on learning and plasticity most robustly, thus identifying distinct biological phenotypes with respect to L-DOPA effects on learning and plasticity. These findings may have clinical applications in post-stroke rehabilitation or the treatment of Parkinson's disease [52]. Indeed, the up-regulation of dopaminergic function by Levodopa may enhance motor memory formation that is crucial for successful rehabilitation of patients with chronic stroke.

After stroke in animal model, tonic neuronal inhibition is increased in the peri-infarct area. The increased tonic inhibition is caused by an impairment in GABA transporter (GAT-3/4) function and is mediated by extrasynaptic GABAA receptors (GABAARs). Treatment with benzodiazepine, an agonist for the α5-subunit-containing extrasynaptic GABAARs, improved recovery of motor function [53].

Treatment with dextroamphetamine combined with physical therapy did not improve recovery of motor function compared with placebo combined with physical therapy as assessed 3 months after hemispheric ischemic stroke. The studied treatment regimen was safe [54].

### 2.4. Drugs to Treat Major Depression in Aging and Disease

The available evidence suggests that affective disorders, such as depression and anxiety, increase risk for accelerated cognitive decline and late-life dementia. Depression is also common in people with Alzheimer's and related dementias prompting to question how affective problems influence cognitive decline, even many years prior to clinical diagnosis of dementia. Indeed, a recent study indicates that cognitive function should be monitored closely in individuals with affective disorders [6].

Suicide attempts may also increase in people diagnosed with dementia. However, the underlying neurobiological pathology remains to a large extent unknown. Several studies have implicated the prefrontal cortex as well as temporal lobe structures in the pathophysiology of affective disorders [55].

Recent advances in transcriptomics identified gene expression changes that were related to inhibitory neurotransmission in spatial learning, neural plasticity, dysregulation of epigenetic mechanisms underlying neurodevelopmental disorders, motivation, addiction and motor disorders, long-term depression, stress response, major depression, and neurovascular unit [56]. Indeed, vascular impairment and subtle patterns of age-related structural brain abnormalities are major contributors to cognitive impairment [57,58].

### 2.4.1. Catecholaminergic Pathway

Neuropsychiatric sequelae, including mood and anxiety disorders, postconcussive syndrome, and personality change, are also common after brain injuries such as TBI and are associated with significant morbidity among survivors [16].

Preclinical and clinical studies involved catecholamine deficits in the etiology of post-stroke depression (PSD) [59]. Therefore, enhancement of central catecholaminergic activity has been considered as a potential treatment strategy for PSD. SSRIs are antidepressants which can be used as therapeutic drugs in the subacute phase of stroke. Indeed, stroke patients who received escitalopram, a widely used SSRI drug, showed improvement in neuropsychological tests assessing memory and executive functions, specifically in verbal and visual memory functions as compared to the placebo [60].

Methylphenidate is a dopamine- and noradrenaline-enhancing agent beneficial for PSD due to its therapeutic effects on cognition, motivation, and mood [59]. Its therapeutic effect is due to DA and noradrenaline reuptake inhibition causing increased concentration of DA and noradrenaline at the synaptic cleft [61]. Methylphenidate is also recommended in the treatment of depression in specific patient sub-groups, such as depression secondary to brain injury [59,62].

### 2.4.2. Anabolic Hormones and Erythropoietin

Gonadal hormones exert potent effects on monoaminergic, cholinergic, and peptidergic pathways as well as neurosteroidogenesis which, in turn, impact normal brain organization and function.

Anabolic steroids, including dehydroepiandosterone, the most potent natural androgen produced by the adrenal gland, are prescription-only medicines used to treat hormonal disturbances in men, used to gain weight after a severe illness, injury, muscle loss in some diseases, or delayed puberty or to treat certain kinds of breast cancer. Thus, anabolic steroids and androgens are prescribed to treat hormonal disturbances in hypogonadism, delayed puberty in boys, or impotence in men [63]. For women, anabolic steroids and androgens are prescribed to treat breast cancer [64]; to inhibit inflammation in endometriosis, a gynecological disorder characterized by the growth of endometrial tissue outside the uterus [65]; to treat osteoporosis; and to treat muscle loss in patients with HIV [66]. Thus, nandrolone, also known as 19-nortestosterone, injected into the muscle or fat tissue is used, although increasingly rarely, in the treatment of anemias, wasting syndrome (cachexia) and as an adjunct therapy in the treatment of senile and postmenopausal osteoporosis or breast cancer. The positive physiological effects of nandrolone esters include muscle growth and appetite stimulation [67]. A recent meta-analysis study suggests that dehydroepiandrosterone (DHEA) may be a more effective therapeutic alternative to the classic drugs used in the treatment of depression [68].

Erythropoietin (EPO) is a glycoprotein hormone mainly produced by the fetal liver and adult kidney and released in response to hypoxia to enhance erythropoiesis. It has been approved since 1989 and is one of the most popular biopharmaceutical products worldwide. It is used in the clinic usually for the treatment of anemia although it has been also tested in neurodegenerative diseases such as AD, PD, and amyotrophic lateral sclerosis as well as in patients with TBI and ischemic stroke [69]. In experimental models of postnatal hyperoxia, erythropoietin had pro-myelinating effects and improved cognition in adolescent and adult rats [70].

At the cellular level, EPO has a variety of effects including angiogenic, antiapoptotic, anti-inflammatory, antioxidant, neurotrophic, and stem cell growth factor. In the brain, EPO is also beneficial for the treatment of motor deficits incurred by ischemic stroke [71] as well as contributing to the preservation of learning and memory abilities [72]. EPO also improves long-term neurological outcome in acute ischemic stroke patients [73,74].

Human growth hormone (hGH) produced by the pituitary gland was approved by the FDA as a safe, effective way to treat conditions associated with short stature due to GH deficiency, Turner Syndrome (a genetic disorder that affects a girl's development) or Prader–Willi syndrome, a genetic disorder causing poor muscle tone. Beauty clinics promote hGH for a variety of purposes, including body rejuvenation, improvement of memory, decrease in fat mass, and increased bone density and muscle mass.

Cognitive impairment including memory, learning, and executive functions is common among stroke survivors and can have a negative impact on quality of life of stroke survivors. GH has been shown to improve cognitive recovery in both rodents and humans. These beneficial effects on cognition are linked to many significant changes within the CNS, including enhanced neurogenesis and vasculogenesis [75–77].

Peripheral administration of GH improved cognitive function in an experimental stroke model. The positive effect was presumably due to increased levels of neurotrophic factors insulin-like growth factor-1 (IGF-1) and vascular endothelial growth factor (VEGF) in peri-infarct regions and periphery [78]. Indeed, several pilot studies in humans indicate a positive outcome after combined GH therapy with specific rehabilitation after stroke and TBI [79–83].

### 3. Chronic Abuse of CNS Stimulants and Anabolic Drugs to Enhance Athletic Performance May Lead to Cognitive Decline and Depressive Behavior

The potential therapeutic use of CNS stimulants and anabolic drugs in poststroke depression, and/or vascular cognitive impairment or vascular dementia is overshadowed by their abuse to enhance athletic performance and overuse (doping) for recreational purposes.

The word "doping" *is attributed to the Dutch* word "doop", *a viscous* opium juice used as a narcotic by the ancient Greeks. Doping to enhance performance has unfortunately become ubiquitous in numerous sports and is commonly used by athletes to improve physical and mental performance.

Athletes use stimulants either to increase performance for fame and financial rewards or as recreational drugs. Reported rates of performance enhancement stimulants use among athletes are variable and range from 5 to 31%. The use of performance-enhancing drugs by children and teenagers has increased tremendously due to the decrease in the age of players in competitive sports and the availability of various stimulants in many forms and shapes. The most common stimulants detected in anti-doping tests include amphetamines, cocaine, ecstasy, methylphenidate and anabolic hormones. Nicotine and caffeine are also frequently used as stimulants but they are not banned in sports.

In this review, we focus on the impact of doping on psychopathological disorders, cognition, and depression. Occasionally, we also refer to chronic use of therapeutic drugs to increase physical performance [84–89]. Indeed, the evidence for the ergogenic (e.g., enhance physical performance) activity of the most stimulants including amphetamines (AMPH), cocaine, and ephedrine, has not been unequivocally proved. A total of 62 stimulants (61 chemical entities) are listed in the World Anti-Doping Agency (WADA) List and prohibited in competition. Athletes may have stimulants in their body for one of three main reasons: (i) inadvertent consumption in a proprietary medicine, (ii) deliberate consumption for misuse as a recreational drug, and (iii) deliberate consumption to enhance performance [90].

The vast majority of stimulants act on the monoaminergic pathways: adrenergic, targeting noradrenaline (sympathomimetic), dopaminergic, targeting DA (dopaminomimetic), and serotonergic targeting serotonin (5-HT, serotoninomimetic). Other psychostimulant drugs include androgenic steroids, hGH, creatine, erythropoietin and AMPH or AMPH derivatives, and beta-hydroxy-beta-methylbutyrate. However, all these stimulants interact with other neural pathways, for example, caffeine, which is an adenosine receptor antagonist, and AMPH and cocaine, which interact with pathways other than those affected by caffeine.

However, most of these substances have serious side effects and pose a serious threat to the health of athletes. Thus, doping may cause serious health-threatening conditions including, infertility, subdural hematomas, liver and kidney dysfunction, peripheral edema, cardiac hypertrophy, myocardial ischemia, thrombosis, and cardiovascular disease [91–95]. The use of cocaine as an illicit substance is implicated as a causative factor for multisystem derangements ranging from an acute crisis to chronic complications.

### 3.1. Drugs That Increase Alertness/Reduce Fatigue

In sporting prolonged (aerobic) exercise-induced fatigue is crucial to the decrease in performance of athletes in competitive events. The culprits are brain serotonin and DA, which are released in response to fatigue as a defense mechanism resulting in reduced intensity of physical exercises [96]. More specifically, the central factors associated with fatigue consist of a number of changes observed in the efferent neurons that alter the recruitment of motor units and appear to be associated with increased serotonergic activity and reduced dopaminergic activity [97–99]. Stimulants that increase alertness/reduce fatigue or activate the cardiovascular system include drugs such as ephedrine, which is available in many over-the-counter pharmacies. Other stimulants that can modify mood such as AMPH, cocaine, and hallucinogenic drugs are available on prescription or illegally. Thus, low doses of a AMPH may enhance physical performance if fatigue adversely affects higher psychomotor activity. Likewise, pseudoephedrine has been suggested to improve high intensity and endurance exercise at high doses.

### 3.2. Amphetamines and Metamphetamines

Amphetamine and amphetamine-related drugs, such as MDMA and methamphetamine (METH), are popular recreational psychostimulants. Recreational use of amphetamines is alarmingly high in those who use drugs for nonmedical purposes and has reached epidemic proportions in the United States and Australasia [100]. AMPH given intravenously or intramuscular or transmucosal or by inhalation is abused by users looking for increased alertness, weight loss, or athletic performance. Although therapeutically used doses are well-tolerated, numerous side effects, such as jitteriness, loss of appetite and psychosis have been reported. Symptoms may last hours to days, and psychosis caused by amphetamine and methamphetamine is characterized by visual hallucinations and persecutory delusions similar to those seen in schizophrenia in terms of symptomatology and pathogenesis and can also cause violent behavior. AMPHs also impair cognition and may actually be the substrate for subsequent psychosis [101]. Short-term use of dextroamphetamine on the 5-choice continuous performance test (5C-CPT) of attention in healthy young adult humans significantly improved 5C-CPT performance [102]. However, in the long run, these drugs, in fact, negatively affect focus, working memory, and sleep quality, creating a vicious cycle.

Chronic use of AMPHs may lead to alterations in the gray and white matter of the brain [101]. Likewise, continuous abuse of AMPH results in altered hippocampal neuronal morphology and disturbances in memory and learning behavior [103].

The nonmedical use of psychostimulants such as Adderall, brings short-term benefits. Long-term use of AMPHs can cause damage to the liver, kidneys, and cardiovascular system. Sustained use of AMPHs can change the structures of the brain involved in cognition, memory, and emotion [104]. For example, using a battery of neuropsychological tests it could be shown that frequent users aged 12–23 performed worse than a group of age- and education level-matched participants [105–108].

AMPHs, methylphenidate, and mephedrone are the most commonly prescribed psychotropic medications in children. Of interest, the synthetic cathinone mephedrone is widely abused by adolescents and young adults. However, little is known of its long-term effects on cognitive function. Indeed, mephedrone seems to induce more harmful effects on cognition than AMPH does during this period of life.

In the recent years, the increase in methamphetamines (MTA) in female abusers has become an emerging problem. However, very little data have been published regarding the effects of prenatal MTA exposure in women. Nevertheless, a case of MTA-related toxicity in a term newborn that has led to early onset of neonatal encephalopathy and liver failure due to its hepatic toxicity has been recently reported. In addition, cranial ultrasonography and magnetic resonance imaging (MRI) showed diffuse white matter damage and two ischemic-hemorrhagic cerebral lesions [109].

### 3.3. Mechanism of Action

In an animal model, the deleterious effects of AMPH and AMPH-related drugs on spatial memory were associated with changes in metalloproteinase 9 levels [110]. Likewise, numerous preclinical studies have demonstrated that AMPH-related drugs may elicit neurotoxic and neuroinflammatory effects [111]. The neurotoxic potentials of MDMA and METH to dopaminergic and serotonergic neurons have been clearly demonstrated in both rodents and non-human primates. In monkeys, AMPH produces long-lasting behavioral changes including hallucinatory-like behaviors and psychomotor depression resembling symptoms of schizophrenia. An examination of regional postmortem levels of DA in tissue from AMPH-naive and AMPH-sensitized monkeys revealed that AMPH sensitization significantly reduced DA turnover in the prefrontal cortex and striatum [112]. At the molecular level, AMPH acts to increase the availability of monoamines DA, NE, and 5-HT in the brain by acting as competitive substrates for the re-uptake of these neurotransmitters by blocking the reuptake transporters of these monoamines. The reuptake transporters normally work by taking up extracellular monoamines to the axoplasm, which is the main mechanism to terminate their activity. Recent evidence suggests that the primary molecular target is the plasma membrane transporter. Upon binding, the monoamine transporter AMPH enters the axoplasm and blocks the re-uptake of monoamine into vesicular monoamine transporter type-2 (VMAT-2). This leads to a massive increase in monoamine concentration at the synaptic cleft [113–115].

A growing body of evidence indicates that AMPH controls gene expression through chromatin modifications. Worrisome, chronic, and acute treatments with AMPH induce epigenetic changes in chromosomes of the users [115]. Therefore, targeting the neural systems and biological pathways underlying these processes may lead to greater success in identifying disease-modifying interventions that would allow us to mitigate the mortality associated with methamphetamine use disorder [116].

### 3.4. Sympathomimetic Stimulants

#### 3.4.1. Noradrenaline and Caffeine

The sympathomimetic (adrenergic) stimulants including noradrenaline and caffeine, centers on their ability to cause persistence of catecholamine neurotransmitters [117]. They may be used to increase alertness, competitiveness, and aggression and are mostly used during training to increase the intensity of the training session [118]. Although caffeine is not considered prohibited substance, a recent study has found that, in well-trained athletes, caffeine ingestion (8 mg/kg body weight), co-ingested with carbohydrates, is responsible for higher rates of post-exercise muscle glycogen resynthesis [45]. Caffeine may also improve the utilization of fatty acids as a fuel source, thereby sparing muscle glycogen [119]. Indeed, the available data show that caffeine is ergogenic and improves physical performance.

#### 3.4.2. Ephedrine

*Ephedra sinica* (EP), a widely used Chinese medicinal plant, acts on part of the sympathetic nervous system (SNS) having similar effects to those of adrenaline. Ephedrine has a long medication history dating back centuries in the world and is commonly used as a nasal decongestant and for weight loss but also for recreational purposes. World Anti-Doping Agency banned ephedrines exceeding the levels (cathine > 5 microg/mL, ephedrine and methylephedrine > 10 microg/mL) of over-the-counter drugs containing nonbanned ephedrines. The effects of EP on the CNS are usually neglected. However, the effects of ephedrines on the brain are still not very well-known because the effects of this drug on the brain usually lead to a diversity of metabolite alteration in different regions. Thus, in an experimental study, three metabolic pathways were impacted by EP in the cortex after administration, including amino acid metabolism, phospholipid metabolism, and amino sugar metabolism [120].

If used for therapeutic purposes with dose control, ephedrine has no potential toxicity. However, if taken in excess amounts, ephedrine is toxic to the brain and can lead to paranoid psychosis, delusions, and hallucinations. Thus, a study conducted on monkeys, showed extensive histological damage, including neuronal degeneration and apoptosis to the prefrontal cortex. FMRI analysis indicated abnormal functional connectivity in the brain regions that perform cognitive control [121]. In humans, a distinctive extrapyramidal syndrome has been observed in intravenous methcathinone (ephedrone) users in Eastern Europe and Russia. Upon admission, the patients reported that the onset of their first gait disturbance occurred after a mean of $5.8 \pm 4.5$ years of methcathinone use. At the time of neurologic evaluation, all 23 patients had gait disturbances and difficulty walking backward, which could be due to methcathinone/Mn toxicity. Indeed, MRI imaging showed disordered myelin sheaths in the white matter, which could be associated with the disorder. Electron microscopic examination of the biopsy also showed frequent abnormalities in mitochondria [122,123].

Mechanism of Action

Ephedrine is a sympathomimetic drug. The principal mechanism of action of ephedrine relies on the indirect stimulation of the adrenergic receptor system by inhibiting neuronal NE reuptake and by displacing more NE from storage vesicles in presynaptic neurons thereby increasing the concentration of NE at the synaptic cleft and allowing more NE to bind to postsynaptic $\alpha$ and $\beta$ receptors which, in turn, increases the production of cyclic AMP, while the alpha-adrenergic effect results from inhibition of adenylcyclase [124].

3.4.3. Cocaine

Daily use of cocaine is rising in adolescents in the USA. However, cocaine and other sympathomimetic drugs have little or no effect on athletic performance [119]. The dramatic increase in cocaine abuse has increased the awareness of the need to understand the effects of cocaine on the brain, especially cognition. Short-term use of cocaine has been associated with increased cognitive performance. This improvement was accompanied by increased activation in the right dorsolateral and inferior frontal cortex, regions considered critical for this cognitive function [125]. The use of cocaine as an illicit substance is implicated as a causative factor for multisystem derangements ranging from an acute crisis to chronic complications. Vasospasm is the proposed mechanism behind adverse events resulting from cocaine abuse, with acute ischemic strokes (AIS) being one of the few [96]. Cocaine use can cause the loss of smell and problems with swallowing and in combination with alcohol may lead to heart attack. In the brain, cocaine abuse was associated with cognitive dysfunction, primarily in working memory, episodic memory, attention, and executive function. The induction of transient psychotic symptoms upon acute administration pf cocaine in healthy volunteers is an important risk factor for the development of psychosis. Cocaine addiction can also cause panic attacks and paranoia [126,127].

Mechanism of Action

Cocaine increases DA neurotransmission by a competitive blockade of monoamine neurotransmitters, including DA, NE, and 5-HT. A blockade of DA reuptake has been closely associated with the reinforcing and addictive properties of cocaine [128]. In the United States, the FDA approved the use of cocaine as a local anesthetic acting as a sodium ($Na^+$) channel blocker [129]. Recent evidence suggests that cocaine increases extracellular DA levels via cocaine-stimulated synthesis of the endocannabinoid 2-arachidonoylglycerol (2-AG) localized in non-synaptic extracellular vesicles (EVs) in the midbrain. Indeed, cocaine causes dissociation of Sig-1R from the ADP-ribosylation factor (ARF6), a G-protein regulating EV trafficking, which in turn leads to activation of the myosin light chain kinase (MLCK) [130]. Intriguingly, in humans, females are more susceptible to the rewarding effects of cocaine than males, an effect that seems to be mediated by estradiol and the metabotropic glutamate receptor 5 (mGluR5) signaling pathway [131].

### 3.5. Anabolic Steroids

Blood transfusions, androstenedione, and DHEA are prohibited in competitive sports [132]. However, misuse of anabolic steroids is common among athletes to enhance performance and prolong endurance but also among bodybuilders and people who feel they need to look muscular and to feel good about themselves. Anabolic-androgens steroids (AAS) were probably among the first AAS to be used as doping agents. For example, nandrolone was much used to improve muscle mass by athletes and sportsmen [133]. Likewise, testosterone precursors such as DHEA, androstenedione, and androstenediol have been heavily marketed as muscle-building nutritional supplements. However, concerns over the safety of these substances led to a ban of over-the-counter selling. Thus, as of January 2005, these hormone precursors cannot be sold without prescription [134]. The anabolic effects including increased muscle mass, improved exercise capacity and energy, of androgen steroids are exacerbated in combination with hGH. In recreational athletes, hGH improves anaerobic sprin capacity [135].

The misuse of anabolic steroids can cause long-term side effects including multiple mental and physical health problems; cerebrovascular complications; liver disease; reproductive organ damage; and severe behavioral disturbances, such as anxiety, aggression, and depressive behavior [136]. In the brain, at the tissue level, supraphysiological doses of androgens have been shown to contribute to impaired cognition, brain aging, neuronal death, and increased inflammation [137]. Indeed, supraphysiologic-dose AAS use has been associated with cognitive decline and brain alterations similar to those found in AD patients [138].

### Mechanism of Action

In animal models, nandrolone increases serotonin levels in brain regions involved in reward-related brain regions [139]. Therefore, chronic use of anabolic steroids has been shown to cause dysfunction of these reward pathways in animals. Specifically, rats given twice daily nandrolone injections for four weeks showed loss of sweet preference indicating reward dysfunction and depressive behavior that was accompanied by reductions in the levels of DA, serotonin, and noradrenaline in the nucleus accumbens, a reward-related brain region. Indeed, in animal model, nandrolone also stimulated the kynurenine pathway, causing increased levels of indoleamine 2,3-dioxygenase and 5-hydroxyindoleacetic acid, a metabolite involved in depressive-like behavior and neurotoxicitys, and *decreased* serotonin levels in the brain [136]. Another study reported that the depressive behavior may be caused by reductions in the DA, serotonin, and noradrenaline contents in the nucleus accumbens [140]. In human subjects, AAS interacts with alpha (1) beta (3) gamma $GABA_A$ receptors and may cause anxiety [141,142].

### 3.6. Benzodiazepines

Benzodiazepines are used to combat insomnia and pain. Benzodiazepine addiction among athletes is a new and growing phenomenon. However, chronic use of benzodiazepine abuse can cause addiction. For example, a young female marathon runner developed lormetazepam addiction after increasing her daily benzodiazepine dosage (18 vials and in total 360 mL) in an attempt to achieve better sleep and enhanced physical performance. She has to be hospitalized for the purpose of benzodiazepine detoxification [143].

Mechanism of Action

After intravenous administration, BZDs quickly distribute to the brain and CNS. Following intramuscular injection, absorption of diazepam is slow whereas intramuscular absorption of injected lorazepam appears to be rapid and complete. Lorazepam is also well absorbed after sublingual administration. BZDs work by acting as a positive modulator on the gamma amino butyric acid GABA$_A$ receptor, a ubiquitous ligand-gated chloride-selective ion channel. Since GABA is inhibitory in nature and controls the extent of neuronal excitability, BZDs produce a calming effect on the brain being sedative, hypnotic, anxiolytic, muscle relaxant, and anticonvulsant. GABA$_A$ receptors properties are at the focus of intense research as their activity is also heavy modulated by BZDs (Valium and Xanax), a class of widely prescribed psychotropic drugs whose sedative properties are used as therapies for anxiety, panic, and insomnia [144].

### 3.7. Erythropoietin

Erythropoietin (EPO) may represent a pharmacological alternative to blood doping by increasing red blood cell mass [119]. Indeed, EPO is beneficial in enhancing haematological parameters, pulmonary measures, maximal power output, and time to exhaustion during maximal physical exercise. However, the literature is inconsistent regarding the performance-enhancing effects of erythropoietin; some studies suggest that it is ergogenic, while other studies suggest that there is no evidence to support the claim [145].

## 4. Conclusions

Currently, restorative therapies that target the brain directly to restore cognitive and motor tasks in aging and disease are available. However, the very same drugs used for therapeutic purposes are employed by athletes as stimulants either to increase performance for fame and financial rewards or as recreational drugs. An overview of use and misuse of brain stimulants is given in Table 1. However, most of these substances have deleterious side effects and pose a serious threat to the health of athletes. Thus, doping may cause serious health-threatening conditions including, infertility, subdural hematomas, liver and kidney dysfunction, peripheral edema, cardiac hypertrophy, myocardial ischemia, thrombosis, and cardiovascular disease. Raising awareness on the health risks of doping in sport for all shall promote an increased awareness for healthy lifestyles across all generations.

**Table 1.** Overview of therapeutic use and abuse of brain stimulants to enhance athletic performance and recreational purposes.

| Drug | Therapeutic Use | Abuse in Sports | Recreational Use | Mechanism of Action | Reference |
|---|---|---|---|---|---|
| Acetycholine receptors; donezepil | Cognitive decline; post-stroke cognitive impairment; AD | NO | NO | Prevent/delay acetylcholine degradation | [20,22,23] |
| Methylphenidate; amphetamine; metham-phetamine; dextroam-phetamine; mephedrone; caffeine; | Cognitive and motivational deficits; poststroke depression; motor recovery after stroke; Parkinson's disease; ADHD | YES; psychoactive substances; increased endurance; increased motor coordination | YES; euphoriant; treat short-term obesity, narcolepsy | Dopamine and noradrenaline reuptake inhibitors; sympathomimetic vasoconstrictor that can raise blood pressure and increase heart rate | [31–34,52–54,59,62,100–104,114,115] |

**Table 1.** *Cont.*

| Drug | Therapeutic Use | Abuse in Sports | Recreational Use | Mechanism of Action | Reference |
|---|---|---|---|---|---|
| Norepinephrine; phenylephrine; ephedrine, methylephedrine; methcathinone | Cardiovascular support | YES; ergogenic; increase muscle glycogen resynthesis | YES | Vasoconstriction; sympathomimetic, inhibition of neuronal NE reuptake; increase cellular cAMP | [36–40,45,118,119,124] |
| Serotonergic drugs L-DOPA; Ephedrine | | YES; increase alertness/reduce fatigue; activate cardiovascular system | NO | Increase serotonergic activity and reduced dopaminergic activity | [97–99] |
| Cocaine, heroin | Anesthetic, pain killer | YES; amphetamine-like properties | YES; euphoriant; pain killer | Increased DA neurotransmission by blockade of DA reuptake | [48–50] |
| Anabolic-androgens steroids (DHEA, androstenedione, androstenediol nandrolone | Hypogonadism; gain weight and muscle loss after severe illness, injury; breast cancer; anemia of renal insufficiency | YES; increase muscle mass, decrease fat mass | YES; treatment of cachexia; loss of muscle loss in elderly; post-menopausal osteoporosis | Anti-catabolic effect by interference with the glucocorticoid receptor | [63–66] |
| Erythropoietin | Treatment of anemia; motor deficits in ischemic stroke | YES; increase red blood cell mass and time to exhaustion during maximal physical exercise | NO | Angiogenic, antiapoptotic, anti-inflammatory, antioxidant, neurotrophic | [71–74] |
| Human growth hormone, hGH | hGH deficiency; Turner Syndrome; Prader–Willi syndrome; cognitive impairment | YES; anabolic effects; increased muscle mass, improved exercise capacity and energy | YES; body rejuvenation, improvement in memory, decrease in fat mass and increased bone density and muscle mass | Increased whole body protein synthesis via IGF-1; lipolytic effects; in the brain, neurotrophic, angiogenic | [78–83,135] |

**Author Contributions:** Conceptualization, D.C., C.-I.C., E.C., and A.P.-W.; writing—original draft preparation, D.C., C.-I.C., E.C., and A.P.-W.; writing—review and editing, D.M.H. and T.R.D.; supervision, I.U. All authors have read and agreed to the published version of the manuscript.

**Funding:** This work was supported by grants from UEFISCDI, iBioStroke, project number 136/2020. under the umbrella of the ERA-NET EuroNanoMed project number 192/2020 (GA #723770 of the EU Horizon 2020 Research and Innovation Programme) to A.P.-W. and grant number PN-III-P4-ID-PCE-2020-059 to A.P.-W.

**Informed Consent Statement:** Not applicable.

**Data Availability Statement:** Data are available from the corresponding author upon reasonable request.

**Acknowledgments:** We acknowledge that this work was supported by grants from UEFISCDI, iBioStroke, project number 136/2020. under the umbrella of the ERA-NET EuroNanoMed project number 192/2020 (GA #723770 of the EU Horizon 2020 Research and Innovation Programme) to A.P.-W. and grant number PN-III-P4-ID-PCE-2020-059 to A.P.-W.

**Conflicts of Interest:** The authors declare no conflict of interest.

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
