# Peer review of "Therapeutic Use and Chronic Abuse of CNS Stimulants and Anabolic Drugs"

_cimb, doi:10.3390/cimb44100333_

Round 1

Reviewer 1 Report

This is a review of potential interest. The concepts the authors discussed may have broad relevance. Below are some of my suggestions that the authors might find worthwhile to consider:

1.      There is a sentence begin with “Conclusions:” at the end of the abstract. This is not a standard format to finish the abstract. The authors could simply say “We conclude …” instead of using “Conclusion:” explicitly.

2.      Page 3, in the second paragraph, there is an error in line break: “moreover, one third of stroke patients are at risk for …”

3.      Similar errors are seen in other places throughout the manuscript. For example in page 4, paragraph 3: “is an alpha-1 adrenergic drug which …”

4.      Page 9, paragraph 4: “Chinise” should be “Chinese”.

5.      I would recommend adding one or two summary figure or table in the manuscript to give an overview about the effects and mechanisms of these different stimulants. This will greatly help the readers to navigate through the manuscript.

6.      The authors emphasized the negative side effects of multiple CNS stimulants in treating various disorders and call for an increase of awareness about these side effects. It would be wonderful if the authors could mention some strategies to help achieve this goal or reduce the negative effects in the usage of these stimulants.

Author Response

Thank you for useful comments which have been all addressed in the revised version. We also added a summarizing Table, as suggested. This article by itself shall improve public awareness of deleterious effects doping on the brain by specialized readers talking to family members, friends, colleagues.

Reviewer 2 Report

In their review article „Therapeutic use and chronic abuse of CNS stimulants and anabolic drugs“ Coliță et al. conflate general psychopharmacological knwoledge with insights about usage of the corresponding substances in ways that they were not primarily intended for. The single substances and their modes of action are outlined adequately. Sequelae of harmful consumption for the nervous system are emphasized througout the whole manuscript. The authors intermingle current literature with essential papers of previous decades. Illustrative clinical examples and cases of detrimental drug administration in other areas of life such as sports enhance comprehensibility and warrant for a fluent read.

Nevertheless, the manuscript is not free of flaws:

In section 3.4.3. Cocaine, the first sentence „Daily use of marijuana is rising in adolescents making cocaine and its derivative tetrahydrocannabinol or THC, the most widely used illicit drug.“ is blatantly erroneus. In the same section it is also wrongly statet: „In the brain, cocaine’s main psychoactive ingredient tetrahydrocannabinol can induce transient psychotic symptoms upon acute administration in healthy volunteers being an important risk factor for the development of psychosis [126,127]“ Again, cocaine is confused with cannabis. To my knowledge this is the only substantial imperfection of the manuscript. All further minor concerns refer to textual aspects and are stated in descending order:

1.) Abstract

„Motor deficits are common after traumatic brain injuries and stroke and affects subjective well-being and are linked with reduced quality of life.“

affect instead of „affects“ as it refers to motor deficits

2.) Section 2.1

„It is estimated that after the age of 65, combined cognitive decline and low mood doubles with every 5 years so that by the age of 85, one in third individuals suffers from Alzheimer’s disease (AD) [3].“

one in three or one third of rather than „one in third“

3.) Section 2.3.1

„Recently, specific acetyl[1]choline receptor targets that boost the positive effects whilst avoiding the negative ones have been identified [20]. The findings, published in Nature Communications, identified specific receptors for the neurotransmitter acetylcholine that re-route information flowing through memory circuits in the hippocampus.“

Reference is already given in the first sentence, naming the journal in the second sentence is redundant?

4.)Section 2.3.2

„Dopamine (3,4-dihydroxyphenethylamine) is a monoamine neurotransmitter synthesized in the brain from L-DOPA that plays a major role in the motivational component of reward-motivated behavior but also movement and cortical plasticity.“

but also in movement instead of „but also movement“

5.) Section 2.3.2

„Methamphetamines, including dextroamphetamine (Adderall), are powerful stimulant of the nervous system that increases the release of DA and NE from presynaptic nerve terminals and is frequently used in the treatment of attention and memory impairment following TBI [31-33].“

stimulants instead of „stimulant“

increase instead of „increases“

6.) section 2.3.2

„The adrenaline agonist, phenylephrine is an alpha-1 adrenergic drug which given intravenously triggers vascular constriction and increases blood pressure.“

which is given instead of „which given“

7.) section 2.3.2

[36]. instead of „[36[.“

8.) section 2.3.2

„Heroin is a potent analgesic, five to ten times more potent morphine.“

more potent than morphine. instead of „more potent morphine.“

9.) section 2.3.3

„Currently, restorative therapies that target the brain directly [46,47].“

That is not a proper sentence. Please rephrase.

10.) section 2.3.3

„Using a combination of electrical stimulation and functional connectivity mapping in a model of stroke, it was shown that administration of an old drug, gabapentin, improves sensory-motor deficits structural and functional plasticity of the corticospinal pathway [51].“

Comma is missing between sensory-motordeficits and structural

11.) section III

„The word “doping” is attributed to the Dutch word “doop” a viscous opium juice used as a narcotic by of the ancient Greeks.“

by the ancient Greeks instead of „by of the ancient Greeks“

12.) section 3.1

„The culprit are brain serotonin and DA which are released in response to fatigue as a defense mechanism resulting in reduced intensity of physical exercises [97].“

culprits instead of „culprit“

13.) section 3.1

„More specifically, the central factors associated with fatigue consist of a number of changes observed in the efferent neurons that alter the recruitment of motor units and appears to be associated with increased serotonergic activity and reduced dopaminergic activity [97-99].“

appear instead of „appears“

14.) section 3.2

„Likewise, continuous abuse of AMPH results in hippocampal neuronal morphology and memory and learning behaviors [103].“

That is not a proper statement. Please rephrase.

15.) section 3.3

„A growing body of evidence indicates AMPH, control gene expression through chro[1]matin modifications.“

AMPH controls instead of „AMPH, control“

16.) section 3.4.1

„They may be used to increase alertness, competitiveness, and aggression are mostly used during training to increase the intensity of the training session [118].“

Aggression and are instead of „aggression are“

17.) section 3.4.1

„Indeed, the available data show that caffeine indeed, is ergogenic and improves physical performance.“

caffeine is instead of „caffeine indeed, is“

18.) section 3.4.2

„Ephedra sinica (EP), a widely used Chinise medicinal plant, acts on part of the sympa[1]thetic nervous system (SNS) having similar effects to those of adrenaline.“

Chinese instead of „Chinise“

19.) section 3.4.2

„However, the toxicity of this amine may become manifest in overdosages ephedrine is toxic to the brain and can lead to paranoid psychosis, delusions, and hallucinations.“

That is not a proper sentence. Please rephrase.

20.) section 3.4.2

„Thus, in a study conducted on monkeys, showed extensive histological damage, including neuronal degeneration and apoptosis to the prefrontal cortex.“

Thus, a study  instead of „Thus, in a study“

21.)section 3.4.2

„BOLD-fMRI analysis indicated abnormal functional connectivity in the brain regions that perform cognitive control [121].“

FMRI rather than „BOLD-fMRI“

22.)section 3.4.2.1

„The principal mechanism of action of ephedrine relies on the indirect stimulation of the adrenergic receptor system by inhibiting neuronal NE reuptake and displacing more NE from storage vesicles in presynaptic neurons thereby increasing the concentration of NE at the synaptic cleft and allowing more NE to bind to postsynaptic α and β receptors which, in turn, increases the productions of cyclic AMP, while the alpha-adrenergic effect result from inhibition of adenylcyclase [124].“

the production of cyclic AMP instead of “ the productions of cyclic AMP“

23.)section 3.4.3

„The use of cocaine as an illicit substance is implicated as a causative factor for multisystem derangements ranging from an acute crisis to chronic complications. Vasospasm is the proposed mechanism behind adverse events resulting from cocaine abuse, acute ischemic strokes (AIS) being one of the few [96]. Cocaine use can cause the loss of smell and problems with swallowing and in com[1]bination with alcohol may lead to heart attack. (…)Cocaine addiction can also cause panic attacks and paranoia.“

That is already stated quite similarly in section III making it redundant at one position:

„The use of cocaine as an illicit substance is implicated as a causative factor for multisystem derangements ranging from an acute crisis to chronic complications. Cocaine use can also cause the loss of smell and problems with swallowing and in combina[1]tion with alcohol may lead to heart attack. Cocaine addiction can cause panic attacks and paranoia. Vasospasm is the proposed mechanism behind adverse events resulting from cocaine abuse, acute ischemic strokes (AIS) being one of the few [96].“

24.) section 3.5.1

„Indeed, in animal model, nandrolone also stimulated the kynurenine pathway causing increased levels of indoleamine 2,3-dioxygenase and 5-hydroxyindole[1]acetic acid and decreased serotonin levels in the brain, a metabolite involved in depressive-like behavior and neurotoxicity [135].“

The subordinate clause „a metabolite involved in depressive-like behavior and neurotoxicity [135]“ should be pasted in another position and not behind serotonin levels to avoid confusion.

25.)section 3.5.1

„Since GABA is inhibitory in nature and control the extent of neuronal excitability, BZDs produce a calming effect on the brain being sedative, hypnotic, anxiolytic, muscle relaxant and anticonvulsant.“

and controls the extent instead of „and control the extent“

26.) „GABAA receptors properties are at the focus of intense research as their activity is also heavy modulated by BZDs (Valium, Xanax), a class of widely prescribed psychotropic drugs those sedative properties are used as therapies for anxiety, panic and insomnia [143]“

psychotropic drugs whose sedative properties instead of„psychotropic drugs those sedative properties“

Author Response

Thank you for useful comments which have been all addressed in the revised version. We also added a summarizing Table, as suggested by one Reviewer.